# PON-P3: Accurate Prediction of Pathogenicity of Amino Acid Substitutions

**DOI:** 10.3390/ijms26052004

**Published:** 2025-02-25

**Authors:** Muhammad Kabir, Saeed Ahmed, Haoyang Zhang, Ignacio Rodríguez-Rodríguez, Seyed Morteza Najibi, Mauno Vihinen

**Affiliations:** Department of Experimental Medical Science, BMC B13, Lund University, SE-22184 Lund, Sweden; muhammad.kabir@med.lu.se (M.K.); saeed.ahmad075@gmail.com (S.A.); haoyang.zhang@med.lu.se (H.Z.); ignacio.rodriguez@ic.uma.es (I.R.-R.); morteza.najibi@med.lu.se (S.M.N.)

**Keywords:** genetic variation, machine learning, variation interpretation, pathogenicity, bioinformatics

## Abstract

Different types of information are combined during variation interpretation. Computational predictors, most often pathogenicity predictors, provide one type of information for this purpose. These tools are based on various kinds of algorithms. Although the American College of Genetics and the Association for Molecular Pathology guidelines classify variants into five categories, practically all pathogenicity predictors provide binary pathogenic/benign predictions. We developed a novel artificial intelligence-based tool, PON-P3, on the basis of a carefully selected training dataset, meticulous feature selection, and optimization. We started with 1526 features describing variations, their sequence and structural context, and parameters for the affected genes and proteins. The final random boosting method was tested and compared with a total of 23 predictors. PON-P3 performed better than recently introduced predictors, which utilize large language models or structural predictions. PON-P3 was better than methods that use evolutionary data alone or in combination with different gene and protein properties. PON-P3 classifies cases into three categories as benign, pathogenic, and variants of uncertain significance (VUSs). When binary test data were used, some metapredictors performed slightly better than PON-P3; however, in real-life situations, with patient data, those methods overpredict both pathogenic and benign cases. We predicted with PON-P3 all possible amino acid substitutions in all human proteins encoded from MANE transcripts. The method was also used to predict all unambiguous VUSs (i.e., without conflicts) in ClinVar. A total of 12.9% were predicted to be pathogenic, and 49.9% were benign.

## 1. Introduction

Next and third generation sequencing methods produce vast amounts of variation data, the interpretation of which has become a bottleneck in the usage of genetic information. Variation interpretation is used to reveal the disease relevance of (genetic) variations of any size. Interpretation guidelines from the American College of Medical Genetics and Genomics and the Association for Molecular Pathology (ACMG/AMP) [1] are followed in many countries and clinical laboratories. The guidelines provide a systematic scheme that summarizes eight types of information. Variants are described with a five-tier classification, which is based on the strength of the information. Computational predictors are widely used for variation interpretation, and many predictors are available [2]. Many of these tools utilize machine learning (ML) algorithms and, more recently, large language models and other advanced approaches.

Despite all these efforts, many variants of uncertain significance (VUSs) exist. Currently, ClinVar [3] contains some 3 million variants, approximately half of which are VUSs. We recently indicated that all variants cannot be classified as pathogenic or benign [4] for many reasons. Pervasive biological heterogeneity, called poikilosis [5], is one reason.

Amino acid substitutions are the most studied variation category, and the largest number of variation interpretation predictors are for these alterations. Many of the prediction methods are for pathogenicity, also called tolerance. These tools predict whether a variant is likely pathogenic without providing any further information. Other methods are more specific for certain mechanisms or effects. The performances of different types of variation interpretation predictors vary widely.

Pathogenicity predictors can be grouped into different categories. Evolutionary data have been widely used (e.g., in SIFT [6], FATHMM [7] and related tools, and PROVEAN [8]). More recently, sequence-based language models have been released (EVE [9] and ESM1b [10]). Other methods utilize multiple data types (PON-P2 [11], PON-All [12], PolyPhen2 [13], and VEST4 [14]). A structure-based predictor (AlphaMissense [15]) and metapredictors (MetaLR and MetaSVM [16], MetaRNN [17], ClinPred [18], and BayesDel [19]) are other types of tools.

For pathogenicity prediction, we developed PON-P [20], which was a metapredictor; PON-P2, which utilized protein and variation features [11]; and PON-All, which is used for amino acid substitutions in any organism [12]. In addition, we have PON-PS for variant severity prediction [21] and dedicated tools for effects, such as stability (PON-Tstab [22]), folding (PON-Fold [23]), and disorder (PON-IDR, submitted).

We present a new highly accurate pathogenicity predictor, PON-P3, which is trained on features that describe genes and proteins, variations and their context, as well as the protein structural details of variations and surrounding regions. To provide a realistic outcome, our tool predicts three categories of variants. In addition to pathogenic and benign variants, there are VUSs. We tested several ML algorithms and trained PON-P3 with a gradient boosting algorithm. Extensive feature selection was used to identify the most significant features among more than 1500 parameters collected for each variant. The datasets for training and testing were systematically selected, including high data quality requirements. Comparison with 23 different types of predictors, a performance assessment, and a benchmark analysis indicated that PON-P3 had high reliability. It is freely available. All possible variations in Matched Annotation between NCBI and EMBL-EBI (MANE) transcript [24]-related protein entries were calculated. The application of this new tool was demonstrated in various studies. We paid special attention to the interpretability of the predictions.

## 2. Results

We compiled a dataset of reliable pathogenic and benign variants from ClinVar and LOVD (Figure 1). From ClinVar, we chose variants with at least a two-star review status (multiple submitters), and from LOVD the effect had to be +/+, i.e., both the submitter and curator agreed on the pathogenicity classification. The data were further filtered to remove duplicates and cases with insertions, deletions, or unknown variations. The variants were subsequently matched to MANE transcripts [24]. For training, we used 11,567 pathogenic and 11,478 benign variants (Table 1). The independent blind test set included 1246 pathogenic and 1335 benign variants. The training and test sets were completely disjointed, and all protein variants were either in the test or training set. The numbers of proteins used for training and blind testing were 3363 and 374, respectively. The datasets cover a wide spectrum of proteins, allowing the generalization of variation effects.

We collected a large number of different types of features to describe variants. In total, we had 1526 features for each variant. The features were at different levels. Protein and gene features were for the entire proteins. These included sequence length, redundancy, and inheritance patterns. The variation features describe the variation and its context, including PSSMs, sequence context, and others. Structural features describe the properties of the protein structure at the variation positions. These included secondary structural elements, accessibility, and other characteristics. The types and numbers of features are shown in Appendix A.

The flowchart in Figure 1 shows the process of developing the new predictor. We performed extensive data collection and curation. Training data were used to implement predictors with all the features. This step was utilized to identify the best-performing algorithm, which was then used to train the final tool. We trained several algorithms and compared their performances.

### 2.1. Choice of the ML Algorithm

We tested four algorithms, including random forests (RF), two versions of boosting algorithms, LightGBM and XGBoost, and a support vector machine (SVM). These algorithms were chosen because of experience in our previous pathogenicity predictors. PON-P2 [11] uses RF and PON-All [12], and several other recent tools are based on boosting algorithms.

RF is an ensemble learning method that creates many decision trees during training and then combines their outputs to increase accuracy while decreasing overfitting [25]. Each tree is trained on a randomly selected portion of the data and features to reduce the association between trees and to improve model adaptability. The final classification prediction is made via majority voting. Decision trees are prone to variance but merging them in a forest reduces the tendency. RF is resilient to overfitting since it averages over varied trees. We found that ntrees = 200 and max_depth = 6 provided the best outcomes.

LightGBM is a gradient boosting framework [26]. It utilizes decision trees to improve performance and effectiveness over classic boosting approaches. It implements a leafwise tree development technique. The model builds trees based on the leaf with the largest loss reduction, instead of the levelwise growth in other methods. The trees are deeper and more accurate with fewer splits. This is an advantage, especially when a large dataset is used. The required amount of memory is smaller than that in many other algorithms; therefore, LightGBM can utilize very large datasets and numerous attributes. The parameters we used in LightGBM were as follows: ntrees = 200; learning_rate = 0.05; max_depth = 12, num_leaves = 64; min_child_samples = 20; subsample = 0.8; and colsample_bytree = 0.8.

XGBoost employs an additive tree model, where each new tree corrects the mistakes of prior trees by minimizing a predetermined loss function [27]. Regularization techniques, such as L1 and L2 penalties, can assist in controlling overfitting. XGBoost has built-in support for missing values and sparse data. Although XGBoost uses efficient approximations, such as second-order gradients and parallelized computations, it can be computationally demanding. The parameters used in XGBoost were ntrees = 200, learning_rate = 0.05, max_depth = 6, min_child_weight = 20, subsample = 0.8, and colsample_bytree = 0.8.

SVM classifies data by finding an optimal hyperplane that maximizes the distance between each class in an N-dimensional space [28]. The kernel function is used to transform the input space into a higher dimensional feature space for efficient classification. We used radial basis function (RBF), which has been widely used in bioinformatic applications, including our protein stability predictor [29]. The parameters used were C = 1.0, kernel = ‘rbf’, degree = 3 (used for polynomial kernel, not relevant for RBF but still present), gamma = ‘scale’, coef0 = 0.0, shrinking = True, tol = 1 × 10^−3^, cache_size = 200, class_weight = None, verbose = False, max_iter = −1, decision_function_shape = ‘ovr’, break_ties = False, and random_state = None.

All the algorithms were trained with a full set of features. The results are shown in Table 2. LightGBM and XGBoost had the highest performance, with practically identical performance. The results for SVM and RF were also good but somewhat lower. The accuracy was 0.924, the MCC was 0.854, and the OPM was 0.81 for LightGBM. The NPV, PPV, sensitivity, and specificity scores were 0.883, 0.975, 0.871, and 0.978, respectively. The accuracy and Δaccuracy are correlated on the normalized data [30]. The lowest value was for sensitivity, and there were more FNs than FPs. The values were normalized since there was a slight imbalance in the positive and negative cases. The numbers of pathogenic variants were normalized to be equal to the number of benign variants. Because of the high performance on all the performance measures, we chose LightGBM to train the final method with selected features.

We optimized the algorithms by testing their performance with different numbers of trees, including 200, 250, or 300 trees (Appendix A). The results for LightGBM and XGboost were identical. We used 250 trees in the predictions in Table 2. The overall measures, accuracy, MCC, and OPM were slightly better when 200 trees were used; however, the differences were very small. We chose 200 trees for the final predictor to make the tool somewhat faster, which was necessary for proteome-wide predictions.

### 2.2. Feature Selection and Method Training

The total number of features collected was 1526. To reduce their number and exclude those that were not informative for the predictions, we performed feature selection by employing a recursive feature elimination algorithm. RF-RFE iteratively trains the predictors and determines the importance scores for each parameter via RF. In each step, feature(s) with the lowest importance were eliminated, and a new model was trained. This process was repeated until a given number of features was obtained. We tested the performance with different numbers of features; see Appendix A. The optimal performance was obtained with 30 features.

GO annotations have been among the most important features, e.g., for PON-P2. However, they are not available for all proteins. To benefit from the GO annotation on the one hand and to predict variations in as many proteins as possible, we trained two predictors, one with GO and another without the GO feature.

The selected features are shown in the beeswarm plot in Figure 2. The feature selection included parameters from all three major classes. The most important features were accessibility, GO, PSSM3, and PSSM2. Four out of the five PSSM scores were selected. There were 18 dipeptide features, four AAindex features, and sequence lengths. The protein structural features included accessibility and low-confidence regions. Regions with low AlphaFold predicted local distance difference test (pLDDT) scores may be disordered and have many interchangeable structures.

The Shapley plot (Figure 2) shows the importance of each feature for pathogenicity (positive values) and benign (negative values) prediction. The color indicates a range of feature values. The blue color represents low values, and the red color represents high values. In the case of binary protein/gene features missing, the feature (indicated by a zero value) is blue, and the existence of the property (e.g., housekeeping protein or haploinsufficiency) is indicated by red. For features with a range of values, the scale indicates the increasing feature value. Positive SHAP values indicate contributions toward pathogenic classification, and negative values indicate contributions toward benign classification.

The features were arranged in descending order of importance. Accessibility was the most important feature; low accessibility scores (blue) were related to pathogenicity and indicated that these important amino acids were mainly buried. The residues with higher accessibility scores (red) had negative SHAP values and were located on the protein surface where they were more accessible. PSSM3 and PSSM2 had opposite correlations. High PSSM2 scores had positive SHAP values, whereas in PSSM3, the situation was the opposite.

The selected AAindex features were an amino acid similarity matrix based on the THREADER force field (DOSZ010103) [31], context-dependent optimal substitution matrices for buried beta (KOSJ950106) and helix (KOSJ950105) [32], and an environment-specific amino acid substitution matrix for alpha residues (OVEJ920102) [33]. DOSZ010103 and KOSJ950106 were also selected features in PON-All.

Both the absence and presence of the dipeptide had positive Shapley Additive exPlanation (SHAP) values, depending on the dipeptide.

For PON-P2, we used site-specific annotations from UniProtKB/SwissProt for functional annotations. This feature was applied after the prediction when the final outcome was processed. For functional sites, the probability of pathogenicity was estimated from the prediction probability and proportion of variations. We tested the applicability in PON-P3; however, the effect on performance was marginal, so this step was omitted. The required annotations appeared only for a small fraction of the variants.

PON-P3 is a LightGBM-based method trained on the 30 most informative features. The performances of the methods are shown in Table 3, both with and without GO annotations. The performance was very close to that of the method with all the features, and that with the GO feature was better. The inclusion of GO annotations increased the accuracy from 0.913 to 0.944 and improved the MCC from 0.828 to 0.888, indicating a significant enhancement in predictive performance. Additionally, coverage increased from 0.698 to 0.749, reflecting the model’s ability to provide confident predictions for a larger proportion of variants.

Further interpretability was obtained by investigating the original and variant amino acid predictor performance per amino acid (Figure 3). The larger the MCC value, the better the amino acid type is predicted. Overall, all the residues presented high performance. The plots for accuracy were practically identical; only the scores were higher since the range for accuracy is from zero to one, whereas the range for MCC is from −1 to 1.

C and W were the best predicted original amino acids with almost perfect MCCs (Figure 3A). Cysteine is a unique residue that can form disulfide bridges essential for protein stability. These sites are vulnerable to alterations. The charged amino acids E, K, R, and polar Y had somewhat lower performances. There were relatively small differences between the amino acids. The situation was very similar for the variants (Figure 3B). Alterations to N and Y, followed by those to S or T, were the most reliably predicted changes. The predictions to C, K, and W had the lowest MCC values, but even those scores were very high.

### 2.3. Benchmarking PON-P3 Performance

To compare the performance of PON-P3 to that of other tools, we obtained predictions for 23 pathogenicity predictors from dbNSFP [34]. These methods represent a broad spectrum of algorithms and prediction types. Several tools are based on evolutionary data from multiple sequence alignments (versions of FATHMM [7,35,36], LIST-S2 [37], MutationAssessor [38], PROVEAN [8], SIFT [39], and SIFT4G [40]) or on language models of sequence families (ESM1b [10] and EVE [9]). PON-P3 belongs, along with CADD [41], DEOGEN2 [42], MCAP [43], PolyPhen2 versions [13], and VEST4 [14], to methods that are based on different types of features. AlphaMissense [15] was the only method based solely on structural information. Structural features were included in addition to PON-P3 to PolyPhen2. The fourth category consisted of six metapredictors that were used as feature predictions from other tools. These tools include BayesDel [19], ClinPred [18], MetaLR and MetaSVM [16], MetaRNN [17], and REVEL [44].

The blind test dataset was used in benchmarking. This dataset contained 2581 variants, of which 1335 were benign and 1246 were pathogenic. These variants appeared in 374 proteins. The variants and proteins were disjointed with the training data, thereby facilitating unbiased benchmarking.

The results in Table 4 were likely somewhat inflated for many of the tools, since cases in our test set have been used to train other methods. Since training datasets were not available for all methods, we could not exclude such cases. For MetaRNN [17], we filtered out cases used for training, but it was not possible with the other metapredictors; therefore, the results are likely somewhat too optimistic for these tools. For an entirely fair comparison, in the case of metapredictors, which combine results from many tools, it would be necessary to exclude all the variants used to train any constituent methods. Since most tools are based on ClinVar data, circularity or data leakage cannot be avoided.

The results indicate widely varied performances. We used the six measures recommended for comprehensive performance assessment [45,46] supplemented with OPM, which was also used to visualize the performance of PON-P3 and other methods (Figure 4). Both versions of PON-P3 were among the tools with the most even distributions for the four measures of sensitivity, specificity, NPV, and PPV. These scores are calculated from half of the data in the confusion matrix (Table 4). The accuracy, MCC, and OPM are overall measures based on all the cells in the confusion matrix.

Figure 4 shows the OPM plot for the performance of PON-P3 along with SIFT, AlphaMissense, and MetaRNN. For clarity, we included only four methods. SIFT is still very widely used, although it has low performance. AlphaMissense is a unique structural features-based tool, and PON-P3 represents single predictors, as well as MetaRNN metapredictors. Figure 4 shows the performance of methods simultaneously based on the six performance measures. When comparing methods, the walls further away from the origin indicate better performance. It is easy to see the relative performance for each of the six measures. The best tools were close to each other. The walls in the cuboid were drawn according to the six performance scores. The larger the cuboid volume for a method, the better its performance. The figure allows easy comparison of tools in relation to each other.

The lowest accuracy in Table 4 was for one of the most widely used methods, CADD (0.671), which also had the lowest scores for MCC (0.447) and OPM (0.371). The results for FATHMM-MKL were only marginally better. Other methods with low performance were M-CAP, LIST-S2, and PolyPhen2 Hvar. Language model-based predictors have been hyped. However, both EVE and Esm1b showed only modest performance.

MetaRNN and ClinPred, both of which are metapredictors, achieved the best performance, followed by PON-P3. The most modern metapredictors can benefit from the constituent predictors and are dependent on the performance of these tools. The other, somewhat older, metapredictors Bayes-Del, MetaLR, MetaSVM, and REVEL, were not among the top performers.

Evolutionary and sequence data were only sufficient for modest performance. Structure-based AlphaMissense had relatively high performance but was not as good as the top performers. The PON-P3 versions were the most reliable among the tools based on many types of features. The best metapredictors may have somewhat overinflated performance since the constituent methods are largely based on ClinVar data, which we used for testing.

### 2.4. Examples of Applications of PON-P3

PON-P3 predictions were made for all possible amino acid substitutions in all the MANE proteins for which features could be obtained, for a total of 18,953 proteins. We made 204,060,990 predictions, of which 33,419,303 (16.4%) were pathogenic, 90,308,318 (44.2%) were benign, and 80,333,369 (39.4%) were VUSs. The results are downloadable from the PON-P3 website.

Figure 5A shows the distributions of the predicted variant outcomes for the 100 proteins with the largest number of cases in our training dataset. The ratios of variation effects varied, as previously noted in the analysis of benchmark datasets [47]. Presenilin-1 (PSEN1) is highly vulnerable, and almost all variants are predicted to be pathogenic (Figure 5A). It is a component of the gamma-secretase complex. Alzheimer’s disease (AD) patients with an inherited form of the disease carry presenilin variants. RP1L1 and ZNF469 are examples from the other end of the spectrum. They contained almost exclusively benign variants. RP1 like 1 is a retinal-specific protein, harmful variations in which cause occult macular dystrophy (OMD). The function of zinc finger protein 469 (ZNF469) is unclear, and it may be a transcription factor. Some variants cause brittle cornea syndrome because of corneal thinning.

We took several measures to understand the bases of the predictions and interpretability of PON-P3. The Shapley plot shows the contributions of the selected features (Figure 2). Interpretability and explainability are gaining attention [48,49]. The more complicated the AI method is, the less interpretable the results are. For example, large language models are practically black boxes, the functioning of which is not possible to explain. In health care, there are increasing demands for explainability of prediction methods [50,51].

Figure 5B–F show an example of the application of PON-P3, along with information that contributes to the interpretability of predictions. We predicted all possible amino acid substitutions in the CD40 ligand CD40LG (Figure 5B), which belongs to the tumor necrosis factor (TNF) superfamily of transmembrane proteins. CD40LG is expressed on the surface of T cells. It regulates B cell function by engaging CD40 on the B cell surface. A defect in this gene results in an inability to undergo immunoglobulin class switch and is associated with hyper-IgM syndrome [52].

Pathogenic variants are concentrated in the C-terminal TNF domain, as well as in the helical region from 19 to 74. Residues 23–46 form a transmembrane helix, vulnerable to substitutions. The C-terminus contains only pathogenic variants or VUSs. Benign variants were most common in the cytoplasmic N-terminus, the second α-helix and the linker connecting to the C-terminal domain.

The results in Figure 5 provide insight into the types of variants. Figure 5C shows the distribution of pathogenic variants in CD40LG. Pathogenicity largely correlates with sequence conservation (Figure 5D) and accessibility of the variant site in the tightly packed TNF domain (Figure 5E).

Figure 5F indicates one property or problem of AlphaFold predictions. The TNF-homology domain at positions 122–261 is well organized; otherwise, the structure is largely elongated. There are two helical regions, 19–74 and 84–105; otherwise, the structure is composed of coils. AlphaFold predicts elongated helices, in this case, with quite high confidence scores (Figure 5F), although the protein is likely much more compactly packed. Even the coils have pLDDT scores over 50, suggesting a well-ordered structure. There are no regions with low pLDDT scores, often indicative of IDRs. AlphaFold may present less well-ordered regions as too unwounded; therefore, residue accessibilities are likely exaggerated in large parts of this protein, which could reduce the usability of accessibility and possibly some other structural parameters as explainable features.

### 2.5. PON-P3 Server and Precalculated Results

PON-P3 is freely available at https://structure-next.med.lu.se/PON-P3/. The user-friendly web server was implemented with Django (version 5.1). The precalculated results are stored in the sqlite3 database.

Variations can be submitted at the genomic, transcript, or protein level. Variants at the nucleotide levels are converted to protein alterations with SeqCAT [53]. Only SNVs that lead to amino acid substitution are allowed. If submitted variants are in proteins for which all features cannot be obtained, a note is provided that predictions are impossible. For example, proteins unique to humans cannot be predicted due to the lack of evolutionary details. If a protein structure is not available, predictions are also missing. This applies to some proteins, especially fibrous proteins, for which we cannot obtain reliable predictions. The predictions for all 19 amino acid substitutions at each position can be obtained by downloading the complete data. Another comment is provided for variants used for the training and testing; there will be a classification along with a note that these cases were not predicted; instead, the original classification is provided.

One can submit several variations at a time in several genes or proteins. The number is not limited; however, we recommend that those with larger numbers of variations download the precalculated predictions. The results are returned by e-mail, which contains two documents: a PDF report and a text file that can be used for further processing.

PON-P3 is entirely MANE-based and MANE-compliant, which means that the submitted variants must be mapped to MANE reference sequences. This is because many features are specific for a variation position and its context. We calculated the outcome of all 19 possible amino acid substitutions in all MANE-related proteins. The results are available for download from the website.

## 3. Discussion

We developed PON-P3, which was extensively tested and found to present very good performance. There are several reasons for this outcome. We used an extensive and diverse set of features and performed a comprehensive feature selection along with selection of the optimal algorithm. We also addressed the limitations of existing predictors by classifying variants into three categories, which are aligned with clinical interpretation guidelines. PON-P3 was comprehensively tested and compared with state-of-the-art methods. It showed excellent performance in both the CV and blind tests. It is likely the best performing pathogenicity predictor available as it classifies variants into three categories. The prediction coverage of the benchmarked methods ranged from 0.544 for PolyPhen2 Hvar to 1.0, with values 0.698 and 0.749 for PON-P3 without and with GO annotations, respectively. All the metapredictors classified at least 99% of the variants as pathogenic or benign. This is beneficial when testing with a binary benchmark but a problem in real life situations.

VUSs constitute the largest group among cases in ClinVar, accounting for 87.9% of amino acid substitutions (October 2024). Thus, binary classifiers make substantial numbers of incorrect predictions. It is possible to somewhat reduce the number of VUSs; predictors are essential for this task.

We used PON-P3 to classify all unambiguous VUSs in ClinVar (cases classified as VUS, excluding conflicting variants). There were 134,942 VUSs, of which 17,384 (12.9%) were predicted to be pathogenic and 67,363 (49.9%) benign. A total of 50,195 (37.2%) VUSs remained. As we have previously shown, there will always be VUSs, and a binary benign/pathogenic dichotomy does not apply to variants [4]. The results are available on the PON-P3 website. In comparison, ESM1b predicted 58% of ClinVar VUSs to be benign and 42% pathogenic [10]. AlphaMissense predicted 57% as benign and 32% as pathogenic of all possible amino acid substitutions [15].

These results indicate that most pathogenicity predictors have a severe overprediction problem, and their true performance is substantially lower than reported. Only PON-P3 and PolyPhen2 have a category for VUSs, in PolyPhen they are named unknown. This is also the reason for the lower coverage of these tools in the benchmark test when variants are predicted to belong to two classes. PON-P3 already handles VUSs successfully; however, it would be of interest to train a predictor with variants in the three categories.

PON-P3 is clearly the best standalone method. Only two metapredictors, ClinPred and MetaRNN, yielded somewhat better results. There is a need for PON-P3. First, metapredictors are dependent on reliable constituent predictors, such as PON-P3. Second, the true performance of the metapredictors and the other tested tools is lower when real cases from individuals are investigated. Apart from PON-P3 and PolyPhen2, all the tools predict variants in two categories, benign and pathogenic, and assume that VUSs do not exist. Third, our recent analysis [4] indicated that there will always be VUSs in relatively high numbers. Therefore, all the benchmarks in the literature and in this study are biased. One should include VUSs in the equation. The reason why we did not do this was that the binary methods cannot be benchmarked on three types of variants. We are confident that in a real-life setting, with variants of all types, PON-P3 has a superior performance and more realistic classification of the cases into three categories, similar to the ACMG/AMP scheme. All the users of variation interpretation tools must remember that the truth of variation effects is not binary; there are three classes.

## 4. Materials and Methods

### 4.1. Datasets

The data for the training and testing of PON-P3 were collected from two reliable sources for genetic variation, ClinVar [54], and LOVD [55]. The keyword “missense” was used to search the ClinVar website. The obtained results were filtered using the following criteria: “single nucleotide” as variation type, “missense” as molecular consequence, and “germline” as classification type. The review status for both benign and pathogenic variations was set to two stars or above. Then, the data were further filtered with proprietary scripts. Duplicates and cases with unknown variations, insertions, or deletions were excluded. Additional pathogenic variants were obtained from the LOVD shared databases. We selected variants with “+/+” in effect, clinical classification as “pathogenic”, cDNA change without symbols “+” or “−” (for intron locations), deletion as “del”, duplication as “dup”, insertion as “ins”, protein change without “?” for unsure classification, “p.0” and “p. (0)” for missing protein, termination as “*”, frameshift as “fs”, and synonymous/silent as “=”. The variants were then aligned with the MANE v1.3 reference sequences [24] and annotated with the Variant Effect Predictor (VEP) [56]. Variants that lacked cDNA, protein, or genomic information and duplicates were eliminated. We obtained 19,530 benign and 12,813 pathogenic variants. To get a balanced dataset, we randomly selected 12,813 benign variants. The dataset is available on the VariBench [57] and PON-P3 website, https://structure-next.med.lu.se/PON-P3/.

### 4.2. Features

Pathogenicity originates from several different mechanisms. To cover this variability, we collected 1526 features describing variations, their sequence and structural context, and parameters for the affected genes and proteins.

#### 4.2.1. Protein and Gene Features

These features are for affected gene/protein and thus are identical for all the variants within the gene/protein. Most of these features were binary, either having the property (marked as 1) or missing it (as 0).

Redundant human protein families were obtained from http://ohnologs.curie.fr (accessed on 3 May 2023). There were 1827 families and 4514 unique duplicated proteins.

A total of 2833 housekeeping proteins were obtained from https://housekeeping.unicamp.br (accessed om 5 May 2023).

A total of 6548 essential/indispensable proteins were obtained from [58].

Haploinsufficient proteins were obtained from the ClinGen Dosage Sensitivity Map (https://www.ncbi.nlm.nih.gov/projects/dbvar/clingen/) (accessed on 16 May 2023) by turning genes “on” and region “off”. After excluding cases with 0 (no evidence), 1 (little evidence), or 2 (emerging evidence) in the HI or TS columns, there were 1499 proteins.

By merging gene lists from three publications, 1156 complete knockout genes were identified. A total of 781 genes were obtained from Narasimhan et al. [59], and 1317 genes were obtained from Saleheen et al. [60]. The data of [61] could not be used as such. We eliminated rows with a sequence MAF > 2%. When two variants had a mean absolute frequency (MAF) less than 2% for both, we included “Number of compound heterozygous carriers” in the variant pair. We retrieved 462 samples, filtering out the NA and 0 cases. A total of 1299 genes were obtained by including the “Observed number of imputed homozygotes” (avoiding 0 and NA). After duplicates were eliminated, the three datasets contained 2633 genes.

Proteins associated with lethality were retrieved from the Mouse Genome Database (accessed on 23 May 2023) [62]. Human orthologs of mouse genes associated with embryonic, prenatal, or perinatal death were identified. A total of 1786 lethality proteins were found.

Protein sequence lenngth was used as a feature.

Inheritance patterns were obtained from the Clinical Genomic Database at https://research.nhgri.nih.gov/CGD/ (accessed on 9 May 2023) [63]. We collected data for 1078 genes with autosomal dominant (AD), autosomal recessive (AR), mixed AD/AR, or X-linked (XL) inheritance.

Protein–protein interaction (PPI) data were downloaded from the STRING database [64] by using 500 as the experimental score threshold. The igraph Python package from https://igraph.org 0.11.8 [65] was used to determine the degree (number of interactions), closeness, betweenness, eigenvector, harmonic, hub score, authority score, page rank, and power centrality of all the proteins.

Gene Ontology (GO) terms were collected from UniProtKB/Swiss-Prot and supplemented with the GOATOOLS python library [66]. Unique GO terms were collected, including all the parent terms of a GO annotation. GO terms were separately collected for pathogenic and neutral variants. The logarithm ratios of the GO frequencies were obtained from$$LR=\sum \log\frac{f\left(P_{i}\right)+1}{f\left(N_{i}\right)+1},$$
where LR is the value for the GO annotations and *f*(*P_i_*) and *f*(*N_i_*) are the frequencies of the ith GO term in the pathogenic and neutral datasets, respectively. To avoid uncertain ratios when GO annotations are missing, we added 1 to all the frequencies. If a protein had no GO annotations, i.e., *LR* = 0, this feature was not considered in the prediction. We trained predictors with and without GO annotations separately.

Protein dipeptide features represent the dipeptides (two consecutive amino acids with a variation within the first position) within protein sequences. There was a total of 400 dipeptide features where the variant position was the first one.

Protein phylogenetic ages were estimated with ProteinHistorian [67].

#### 4.2.2. Variation Features

The features in this category describe different aspects of the variation and variation position.

For the variation type feature, we used two matrices to describe amino acid substitutions. The 20 × 20 matrix yielded 400 features. The dimensions are denoted as original and variant residues. An additional 36 features in a 6 × 6 matrix were associated with the chemical and physical characteristics of amino acid alterations. The six residue groups were hydrophobic (V, I, L, F, M, W, Y, and C), negatively charged (D and E), positively charged (R, K, and H), conformational (G and P), polar (N, Q, and S), and other (A and T) amino acids.

The neighborhood features were used for the sequence context of the variation locations. The 20-dimensional vector of neighborhood residues counts the instances of amino acids within a window of 23 positions, i.e., 11 positions before and after the variation site [68]. Another vector indicates the frequencies of five different categories of amino acids within the neighborhood of 23 positions for nonpolar, polar, charged, positively charged, and negatively charged residues.

One feature, called variation in the first position, is for the first amino acid in the sequence. Initiation codon variations prevent protein production.

Amino acid properties included propensities calculated based on AAindex [69], electronic charge index (ECI), and isotropic surface area (ISA). From the AAIndex, we selected 617 complete amino acid propensity scales. ECI represents the net electric charge of an amino acid, which affects interactions with the environment. The ISA of amino acids is a measure of the accessible surface area that is available for interaction with surrounding molecules. The differences between the scores for the original and variant amino acid were computed. ProtDCal [70] was used to calculate three features. W(U) is the number of water molecules close to a residue in an unfolded state. Gw(U) is the free energy contribution from the entropy of the first shell of water molecules in an unfolded state. Gs(U) indicates the interfacial free energy contribution of an unfolded state.

The residue feature refers to the relative position of a variant in the protein sequence.

A position-specific scoring matrix (PSSM) was used to determine sequence conservation. We extracted all mammalian, rodent, and vertebrate sequences from the UniProt_T database and stored them in a database. Each MANE sequence was compared to the database with BLAST, version 2.12.0+ [71]. The e-value limit was set at 0.001, and the maximum number of target sequences was 20,000. The sequence identity criterion was set at 30 for minimum and 80 for maximum values. The sequences were clustered via CD-HIT [72] with parameters of -c 0.8 and -aL 0.8 for the sequence identity threshold and alignment coverage for the longest sequence, respectively. Representatives for each cluster were identified from the outputs. The sequence was eliminated if there were four or fewer sequences in a cluster. If a MANE protein was suggested to represent a cluster, another protein was selected as a representative. The sequences were gathered from the databases via the makeblast technique. The conservation scores for each MANE protein sequence were determined via PSIBLAST [71] with num_alignments = 50,000 and num_iterations = 3.

Five features were obtained from the alignment information in the PSSMs. PSSM1 has conservation scores for each amino acid substitution. Positive numbers indicate that identical or similar sequences are aligned, whereas negative values originate from a nonconserved alignment position.

PSSM2 provides weighted observed percentages that were rounded down. The information content per position (PSSM3) indicates the degree of conservation of positions across the aligned sequences. PSSM4 is the relative weight of gapless real matches to pseudocounts. It shows the proportion of the weight of actually observed matches compared with the pseudocounts. PSSM5 is a new score to describe the contribution of the sequence position to the alignment. It was calculated as follows: the total number of sequences matching with the position was divided by the total number of sequences in the alignment. PSSM1 and PSSM2 have scores for each amino acid substitution. PSSM3 to PSSM5 summarize data over all the substitutions and have only single values.

Information about a pseudogene(s) for the investigated genes was collected from the t2t human genome at the NCBI website, https://www.ncbi.nlm.nih.gov/datasets/gene/taxon/9606/?gene_type=pseudogenes (accessed on 18 January 2024). There were pseudogenes for 3433 genes.

The locations of the variation positions within protein domains were obtained from the InterPro database [73].

To determine whether variations were located within the protein repeat segment, we used T-REKS [74] to search for each MANE sequence.

#### 4.2.3. Structural Features

Structural features describe the characteristics of variation positions within protein three-dimensional structures.

The protein structures, were predicted with either AlphaFold2 [75] or AlphaFold 3 [76] and obtained from the AlphaFold database, CHESS 3 [77], or straight from the predictor. AlphaFold utilizes experimental structures when available. Experimental structures rarely cover the entire protein chain; therefore, models are important.

The coordinates were used to determine the locations of secondary structural elements with STRIDE [78]. The secondary structural features included α-helices, β-sheets, π-helices, 3_10_ helices, isolated β-bridges in two categories, turns/coils, and low confidence (vlow) regions.

The solvent-accessible surface areas (SASAs) of the original amino acids were computed via the FreeSASA Python package [79].

The Human Transmembrane Proteome database provides information for human transmembrane proteins (accessed on 18 January 2024) [80]. To determine positions within human transmembrane (TM) regions, we acquired 5467 TM areas and mapped them to the MANE reference sequences.

Intrinsically disordered protein regions (IDRs) were obtained from the DisProt website (accessed on 18 January 2024) [81]. Duplicates were removed, and directly adjacent IDRs were merged. Subsequently, the sequences were mapped to the MANE reference sequences. A total of 1706 proteins contained one or several IDRs.

### 4.3. Functional Annotations

Annotations for amino acid positions were obtained from UniProtKB/Swiss-Prot and PDB. The annotations included “SIGNAL”, “CROSSLNK”, “ACT_SITE”, “BINDING”, “MOD_RES”, “TRANSMEM”, “INTRAMEM”, “NON_STD”, “LIPID”, “CARBOHYD”, and “DISULFID”. The variations that occur at such sites were identified. The distributions of the annotations in the pathogenic and neutral datasets were calculated.

For variants in functional sites, the probability of pathogenicity was estimated from the prediction probability and proportion of variations annotated occurring in functional sites in pathogenic class as follows:$$P_{c}\left(p\right)=P_{a}\left(p\right)+P_{rf}\left(p\right)-P_{a}\left(p\right)\times P_{rf}\left(p\right)$$
where P_c_(p) is the combined probability of pathogenicity for the variation, P_a_(p) is the probability for the pathogenic variant derived from the proportion of pathogenic variations in the training dataset for the annotation type, and P_rf_(p) is the predicted probability of pathogenicity of the variation.

### 4.4. Feature Selection

We had a very large number of features for a dataset with a large but still limited number of cases. Feature selection was used to find the most important features to train the final predictor. First, we preprocessed the features by removing features with a standard deviation of zero, as they do not provide discriminatory information. A total of 253 such features were removed. Then, we calculated the Pearson correlation coefficient between all possible pairs of the remaining features. We removed highly correlated features to reduce multicollinearity and retained only one feature from pairs of features with a Pearson correlation coefficient greater than 0.8, indicative of a strong linear relationship. A total of 449 features were filtered, leaving 824 features. Random forest recursive feature elimination (RF-RFE) [82], a wrapper-based feature selection approach, was then used to identify the most important features. RF-RFE used the RF training model to determine the weights of each feature in the feature vector. The features were arranged in order of their weight. The lowest-ranked feature(s) were removed. Iterative elimination was continued until the highest prediction rates were attained.

### 4.5. Performance Assessment

High confidence predictions in PON-P3 were identified with a probabilistic method and using several parallel predictors, with 200 trees in the final version. It is impossible to determine the probability distribution function of bootstrap probabilities; therefore, we utilized Chebyshev’s inequality, which applies to any distribution. For a random variable *X* with mean *μ* and standard deviation *σ*, Chebyshev’s inequality guarantees that at least 1 *−* (1/*k*^2^) prediction values lie within *k* standard deviations from the mean$$P\left(\mu -k\sigma <X<\mu +k\sigma \geq 1-\frac{1}{k^{2}}\right).$$

We used 0.95 as the cutoff value. If 1 *−* (1/*k*^2^) was ≥0.95, and if the range for *μ ± kσ* did not include 0.5, the prediction was considered reliable and classified either as pathogenic or neutral. Otherwise, the variation belongs to the VUS category.

### 4.6. Performance Evaluation

Several measures were used to assess the performance of PON-P3. We assessed the tool performance according to published recommendations for performance assessment [45,46]. The measures included positive predictive value (PPV) and negative predictive value (NPV), sensitivity, specificity, accuracy, the Matthews correlation coefficient (MCC), normalized MCC, and the overall performance measure (OPM) as follows$$PPV=\frac{TP}{TP+FP}$$$$NPV=\frac{TN}{TN+FN}$$$$Sensitivity=\frac{TP}{TP+FN}$$$$Specificity=\frac{TN}{TN+FP}$$$$Accuracy=\frac{TP+TN}{TP+TN+FP+FN}$$$$Random\;accuracy=\frac{\left(TP+FN\right)\left(TP+FP\right)+(TN+FN)(TN+FP)}{{\left(TP+TN+FP+FN\right)}^{2}}$$∆Accuracy=Accuracy−Random accuracy$$MCC=\frac{\left(TP\times TN\right)-\left(FP\times FN\right)}{\sqrt{\left(TP+FN\right)\times \left(TP+FP\right)\times \left(TN+FN\right)\times \left(TN+FP\right)}}$$$$nMCC=\frac{1+MCC}{2}$$$$OPM=\frac{\left(PPV+NPV)(Sensitivity+Specificity)(Accuracy+nMCC\right)}{8}.$$

TP and TN are correctly predicted pathogenic and neutral cases, respectively, and FN and FP are the numbers of incorrect predictions for pathogenic and neutral cases, respectively.

Coverage measures the ratio of the number of predicted cases among all the instances. X indicates the number of cases classified as harmful or neutral, and Y is the total number of test variants,$$\mathrm{Coverage}\;=\frac{X}{Y}.$$

## Figures and Tables

**Figure 1 ijms-26-02004-f001:**
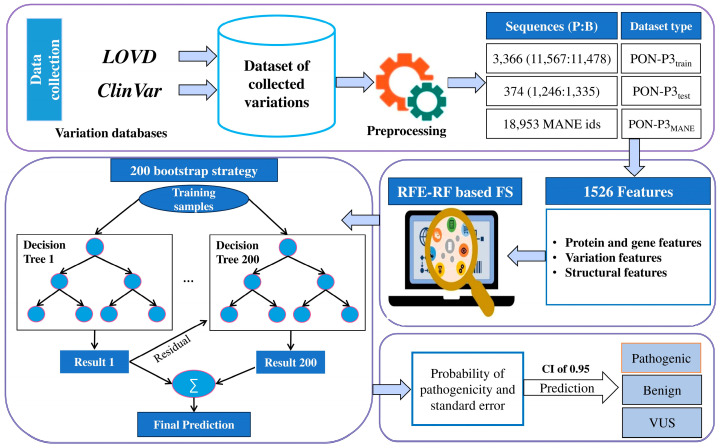
Flowchart of the method development process. We collected training and test datasets from ClinVar and LOVD. Then, the features were calculated and used in feature selection. We used a bootstrap strategy to run 200 parallel predictions combined into the final prediction. PON-P3 classifies variants as benign, pathogenic, or VUS.

**Figure 2 ijms-26-02004-f002:**
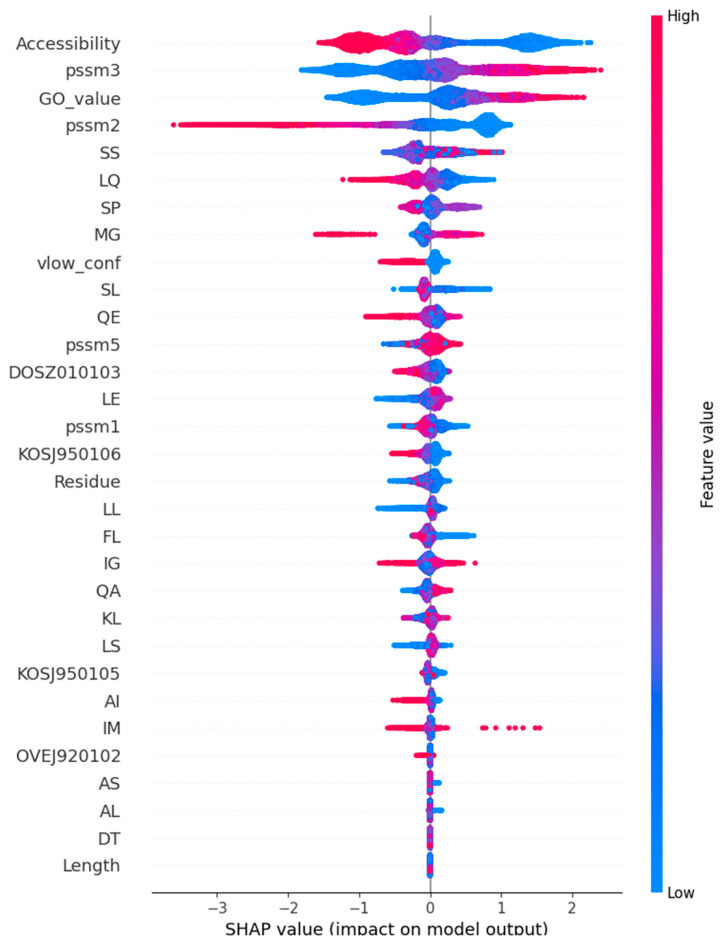
Shapley plot for the 30 selected features organized in descending order of importance. The feature values are colored based on their value, ranging from blue to red. The SHAP value indicates the impact of the features for both positive and negative predictions. A positive result in this context indicates a pathogenic, negative, benign phenotype. Abbreviation: pssm, position-specific scoring matrix.

**Figure 3 ijms-26-02004-f003:**
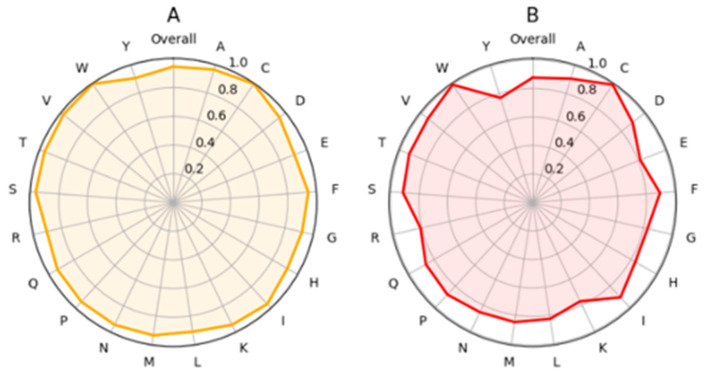
Amino acid-specific performance of PON-P3. MCC values were calculated separately for (**A**) original and (**B**) variant amino acids. The spider plots show the differences between performances for each residue type.

**Figure 4 ijms-26-02004-f004:**
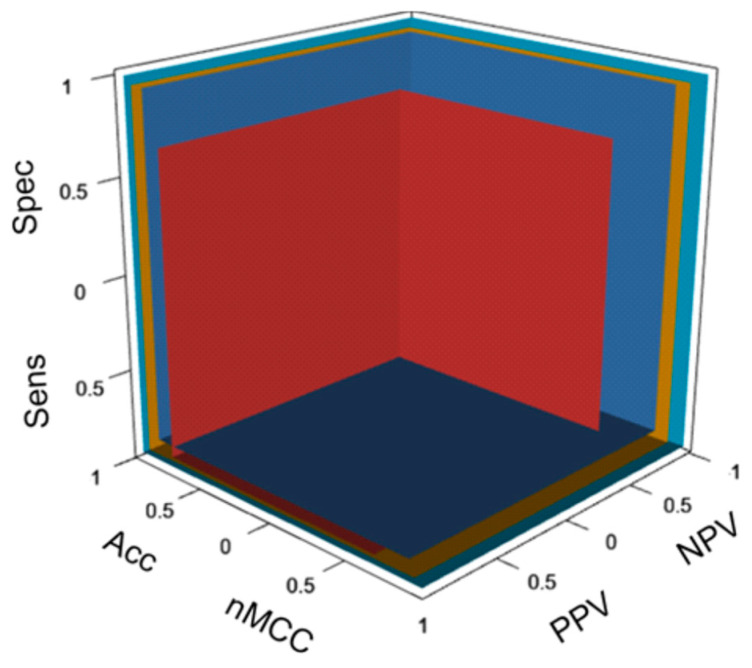
Comparison of the six central performance measures in the OPM cuboid for four different types of predictors. The walls are drawn according to the performance scores. The larger the volume of the cuboid is, the better the performance. The results for SIFT are shown in red, for AlphaMissense in dark blue, for PON-P3 in gold, and for MetaRNN in pale blue.

**Figure 5 ijms-26-02004-f005:**
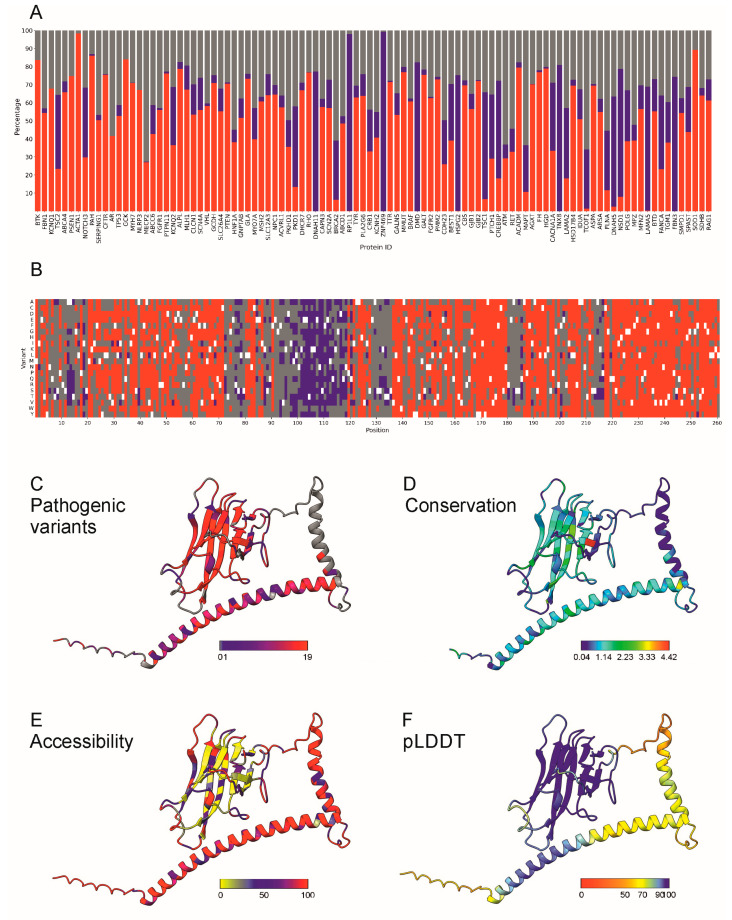
(**A**) Comparison of the predicted categories for the 100 proteins that contained the largest number of variations in the training dataset. Pathogenic variants are in red, benign ones in blue, and VUSs in gray. The graph indicates the percentages of the predicted variant types, with (**B**) showing predictions of all possible amino acid substitutions in CD40LG. The original amino acids are white, and the benign variants are blue, the pathogenic in red, and the VUSs are grey. Visualization of site-specific (**C**) numbers of predicted pathogenic variants, (**D**) sequence conservation according to PSSM3, (**E**) accessibility of the original amino acid, and (**F**) pLDDT score for the local confidence of the AlphaFold predictions. The scales indicate the ranges of scores.

**Table 1 ijms-26-02004-t001:** Numbers of variations and proteins in the training and test datasets.

	Training and CV		Blind Test Set		Total	
	Variations	Proteins	Variations	Proteins	Variations	Proteins
Pathogenic	11,567	3366	1246	374	12,813	3740
Benign	11,478		1335		12,813	
Total	23,045		2581		25,626	

**Table 2 ijms-26-02004-t002:** Comparison of method performance in terms of 10-fold CV when all the features are used for training.

Parameter	LightGBM ^a^	RF	SVM	XGBoost
Accuracy	0.933 (0.924)	0.892 (0.889)	0.926 (0.925)	0.933 (0.925)
ΔAccuracy	0.424	0.389	0.425	0.425
MCC	0.864 (0.854)	0.785 (0.781)	0.852 (0.851)	0.864 (0.855)
PPV	0.966 (0.975)	0.917 (0.928)	0.932 (0.939)	0.966 (0.975)
NPV	0.913 (0.883)	0.874 (0.857)	0.921 (0.913)	0.913 (0.885)
Sensitivity	0.871 (0.871)	0.844 (0.844)	0.91 (0.91)	0.873 (0.873)
Specificity	0.978 (0.978)	0.934 (0.934)	0.94 (0.94)	0.977 (0.977)
OPM	0.81 (0.81)	0.711 (0.711)	0.794 (0.794)	0.81 (0.81)
Pathogenic as unknown	4996	3971	4140	4871
Neutral as unknown	2.388	2631	3191	2362
TP	5722 (7916)	6410 (7466)	6758 (7541)	5845 (7957)
TN	8889	8264	7793	8910
FP	201	583	494	206
FN	849 (1174)	1186 (1381)	669 (746)	851 (1159)
Coverage	0.68	0.714	0.682 (0.693)	0.686

^a^ Numbers in brackets are for normalized cases. For each tool, the numbers of pathogenic variants were normalized to be equal to the number of benign variants. OPM, overall performance measure.

**Table 3 ijms-26-02004-t003:** Performance on the blind test set when all features are used and for PON-P3 with (w) or without (wo) the GO feature.

Parameter	All Features	woGO	wGO
Accuracy	0.92 (0.92) ^a^	0.918 (0.913)	0.945 (0.944)
ΔAccuracy	0.420	0.413	0.444
MCC	0.84 (0.84)	0.833 (0.828)	0.889 (0.888)
PPV	0.919 (0.926)	0.937 (0.95)	0.947 (0.953)
NPV	0.922 (0.914)	0.905 (0.881)	0.942 (0.935)
Sensitivity	0.913 (0.913)	0.871 (0.871)	0.934 (0.934)
Specificity	0.927 (0.927)	0.955 (0.955)	0.954 (0.954)
OPM	0.779 (0.779)	0.771 (0.771)	0.843 (0.843)
Pathogenic as unknown	295	458	340
Neutral as unknown	281	322	308
TP	868 (962)	686 (882)	846 (959)
TN	977	967	980
FP	77	46	47
FN	83 (92)	102 (131)	60 (68)
Coverage	0.777	0.698	0.749

^a^ Numbers in brackets are for normalized cases. For each tool, the numbers of pathogenic variants were normalized to be equal to the number of benign variants. OPM, overall performance measure.

**Table 4 ijms-26-02004-t004:** Blind test performance of PON-P3 compared with other predictors ^a^.

Predictor	TP	TN	FP	FN	PPV	NPV	Sensitivity	Specificity	ACC	MCC	OPM	Coverage
Evolutionary data/sequence information-based predictors												
ESM1b	945 (1082)	970	240	112 (128)	0.797 (0.818)	0.896 (0.883)	0.894 (0.894)	0.802 (0.802)	0.845 (0.848)	0.695 (0.699)	0.607 (0.612)	0.966
EVE	623 (439)	549	48	225 (158)	0.928 (0.901)	0.709 (0.777)	0.735 (0.735)	0.92 (0.92)	0.811 (0.827)	0.646 (0.666)	0.553 (0.576)	1.0
FATHMM	851 (965)	870	284	167 (189)	0.75 (0.773)	0.839 (0.822)	0.836 (0.836)	0.754 (0.754)	0.792 (0.795)	0.589 (0.592)	0.501 (0.504)	0.925
FATHMM-MKL	1059 (1242)	546	717	18 (21)	0.596 (0.634)	0.968 (0.963)	0.983 (0.983)	0.432 (0.432)	0.686 (0.708)	0.484 (0.498)	0.395 (0.412)	0.997
FATHMM-XF	967 (1108)	819	355	58 (66)	0.731 (0.757)	0.934 (0.925)	0.943 (0.944)	0.698 (0.698)	0.812 (0.821)	0.653 (0.662)	0.56 (0.57)	0.937
LIST-S2	841 (936)	647	378	80 (89)	0.69 (0.712)	0.89 (0.879)	0.913 (0.913)	0.631 (0.631)	0.765 (0.772)	0.562 (0.567)	0.471 (0.477)	1.0
MutationAssessor	846 (966)	827	278	122 (139)	0.753 (0.777)	0.871 (0.856)	0.874 (0.874)	0.748 (0.748)	0.807 (0.811)	0.623 (0.628)	0.533 (0.538)	0.883
PROVEAN	842 (950)	937	210	175 (197)	0.8 (0.819)	0.843 (0.826)	0.828 (0.828)	0.817 (0.817)	0.822 (0.823)	0.644 (0.645)	0.555 (0.557)	0.922
SIFT	947 (1067)	769	377	70 (79)	0.715 (0.739)	0.917 (0.907)	0.931 (0.931)	0.671 (0.671)	0.793 (0.801)	0.617 (0.624)	0.523 (0.532)	0.922
SIFT 4G	918 (1028)	890	268	116 (130)	0.774 (0.793)	0.885 (0.873)	0.888 (0.888)	0.769 (0.769)	0.825 (0.828)	0.658 (0.661)	0.568 (0.572)	0.934
Multiple features utilizing predictors												
CADD	1069 (1261)	443	827	8 (9)	0.564 (0.604)	0.982 (0.98)	0.993 (0.993)	0.349 (0.349)	0.644 (0.671)	0.432 (0.447)	0.353 (0.371)	1.0
DEOGEN2	796 (935)	993	136	165 (194)	0.854 (0.873)	0.858 (0.837)	0.828 (0.828)	0.88 (0.88)	0.856 (0.854)	0.71 (0.709)	0.625 (0.624)	0.89
M-CAP	1064 (457)	219	242	10 (4)	0.815 (0.654)	0.956 (0.982)	0.991 (0.991)	0.475 (0.475)	0.836 (0.733)	0.599 (0.545)	0.531 (0.451)	1.0
PolyPhen2 Hvar	119 (522)	884	155	118 (517)	0.434 (0.771)	0.882 (0.631)	0.502 (0.502)	0.851 (0.851)	0.786 (0.677)	0.334 (0.377)	0.323 (0.324)	0.544
PolyPhen2 Hdiv	865 (910)	749	231	67 (70)	0.789 (0.798)	0.918 (0.915)	0.928 (0.929)	0.764 (0.764)	0.844 (0.846)	0.700 (0.702)	0.612 (0.615)	0.933
PON-P3 woGO	686 (882)	967	46	102 (131)	0.937 (0.95)	0.905 (0.881)	0.871 (0.871)	0.955 (0.955)	0.918 (0.913)	0.833 (0.828)	0.771 (0.771)	0.698
PON-P3 wGO	840 (961)	981	50	61 (70)	0.944 (0.951)	0.941 (0.933)	0.932 (0.932)	0.952 (0.952)	0.943 (0.942)	0.885 (0.884)	0.837 (0.837)	0.749
VEST4	997 (1140)	947	232	34 (39)	0.811 (0.831)	0.965 (0.96)	0.967 (0.967)	0.803 (0.803)	0.88 (0.885)	0.773 (0.781)	0.694 (0.704)	1.0
Structural data-based predictor												
Alpha Missense	827 (1005)	1,071	72	114 (138)	0.92 (0.933)	0.904 (0.886)	0.879 (0.879)	0.937 (0.937)	0.911 (0.908)	0.82 (0.818)	0.754 (0.75)	0.888
Metapredictors												
BayesDel	1030 (1208)	1,121	142	47 (55)	0.879 (0.895)	0.96 (0.953)	0.956 (0.956)	0.888 (0.888)	0.919 (0.922)	0.841 (0.846)	0.78 (0.786)	0.997
ClinPred	1018 (1185)	1,244	8	58 (67)	0.992 (0.993)	0.955 (0.949)	0.946 (0.946)	0.994 (0.994)	0.972 (0.97)	0.944 (0.941)	0.918 (0.914)	0.992
MetaLR	965 (1120)	1,092	157	111 (129)	0.86 (0.877)	0.908 (0.894)	0.897 (0.897)	0.874 (0.874)	0.885 (0.886)	0.769 (0.771)	0.693 (0.694)	0.991
MetaRNN	643 (702)	734	4	33 (36)	0.994 (0.994)	0.957 (0.953)	0.951 (0.951)	0.995 (0.995)	0.974 (0.973)	0.948 (0.947)	0.924 (0.922)	1.0
MetaSVM	970 (1126)	1,129	120	106 (123)	0.89 (0.904)	0.914 (0.902)	0.901 (0.902)	0.904 (0.904)	0.903 (0.903)	0.805 (0.805)	0.735 (0.736)	0.991
REVEL	825 (1128)	1,001	160	24 (33)	0.838 (0.876)	0.977 (0.968)	0.972 (0.972)	0.862 (0.862)	0.908 (0.917)	0.824 (0.839)	0.757 (0.776)	1.0

^a^ Numbers in brackets are for normalized cases. For each tool, the numbers of pathogenic variants were normalized to be equal to the number of benign variants. OPM, overall performance measure.

## Data Availability

PON-P3 is freely available at https://structure-next.med.lu.se/PON-P3/.

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
