# Peer review of "PON-P3: Accurate Prediction of Pathogenicity of Amino Acid Substitutions"

_ijms, 2025, doi:10.3390/ijms26052004_

Round 1

Reviewer 1 Report

Comments and Suggestions for Authors

I enjoyed reading the manuscript; it was a pleasure. However, there are several issues/questions that need to be addressed:
Major Comments:

1) Remarks related to the web service:

a) https://structure-next.med.lu.se/PON-P3/input/
There is some input validation, but it is still insufficient:
NP_000008.1
Y382W
S412N
A412N
W412N
This input is still processed, even though it is evident that 412 is neither A nor W. The email response does not explain what happened with these invalid inputs; it simply excludes them without notification.

b) Why do you require email addresses? After clicking "Start prediction," the server seems to freeze for 5–10 seconds before displaying "Email sent successfully! Check your email," and the results are sent immediately. This indicates that the method is fast enough to return results directly on the webpage, such as by presenting the content of the prediction_result.pdf.

c) The output of the service is unclear. For example:
https://structure-next.med.lu.se/PON-P3/input_transcript/
Given the following input:
NM_000016.6:c.613G>C  
NM_000016.6:c.613G>A  
NM_000016.6:c.613G>T  
NM_000016.6:c.613G>G  
NM_000020.3:c.1121G>A  
NM_000020.3:c.1121G>C  
NM_000020.3:c.1121G>T  
NM_000020.3:c.1121G>G  
NM_000021.4:c.506C>G  
NM_000021.4:c.506C>A  
NM_000021.4:c.506C>C  
NM_000021.4:c.506C>T  

The output was:
transcript_variation,refseq_ids,variation_ids,meanProb,stdProb,pred_label,comments  
NM_000016.6:c.613G>C,NP_000007.1,A205P,0.984,0.009,Pathogenic,This variation was used to train PON-P3.  
NM_000020.3:c.1121G>A,NP_000011.2,R374Q,0.98,0.011,Pathogenic,This variation was used to train PON-P3.  
NM_000021.4:c.506C>T,NP_000012.1,S169L,0.699,0.101,Unknown,This variation was used to train PON-P3.  
NM_000021.4:c.506C>G,NP_000012.1,S169*,,,,There is no data for this variation  
NM_000020.3:c.1121G>C,NP_000011.2,R374P,0.999,0.001,Pathogenic,  
NM_000020.3:c.1121G>T,NP_000011.2,R374L,0.998,0.001,Pathogenic,  
NM_000020.3:c.1121G>G,NP_000011.2,R374=,,,,There is no data for this variation  
NM_000021.4:c.506C>A,NP_000012.1,S169*,,,,There is no data for this variation  
NM_000021.4:c.506C>C,NP_000012.1,S169=,,,,There is no data for this variation  
NM_000016.6:c.613G>A,NP_000007.1,A205T,0.99,0.006,Pathogenic,  
NM_000016.6:c.613G>T,NP_000007.1,A205S,0.986,0.009,Pathogenic,  
NM_000016.6:c.613G>G,NP_000007.1,A205=,,,,There is no data for this variation  

d) Why are the results randomly reordered? For example, the following input:
NM_000016.6:c.613G>C  
NM_000020.3:c.1121G>A  
NM_000021.4:c.506C>T  
results in outputs that appear to lack order. It also seems that if there is no conflict, everything is labeled as "Pathogenic."

Similar improvements could be applied to:
https://structure-next.med.lu.se/PON-P3/input_transcript/
https://structure-next.med.lu.se/PON-P3/input_genomic/

Additionally, the email subject lines and content are identical for all three sub-services. Customize these to be more informative, as the current setup makes emails appear generic or even like spam.

2) Since "PON-P3 is entirely MANE-based and MANE-compliant," it is essential to state the exact version of MANE used. This critical information is missing from both the website and the manuscript. For example, the available MANE releases include:

release_0.5/  
release_1.0/  
release_1.4/  

If the version used is below 1.0, it would be advisable to update PON-P3 to the latest MANE protein list.

3) Please provide the predictions for both the test and training datasets for PON-P3 and all other methods to which you compare PON-P3 (as presented in Table 4). Without this, the correctness of the work cannot be validated. Additionally, providing the scripts for calculating the statistics would save me a time and improve transparency.

4) Even better, provide the feature assignments for all variants.

5) Given the dataset size and the number of features used, it would be worth testing one additional algorithm (SVM with RBF & grid search) alongside RF, XGBoost, and LightGBM.

Minor Comments:
- Correct the URL in the abstract (line 33):
http://structure-next.lu.se/PON-P3 → https://structure-next.med.lu.se/PON-P3/ (line 673 is already correct).
- Specify the exact license. The information at https://structure-next.med.lu.se/PON-P3/disclaimer/ is insufficient. Explicitly state whether it is MIT, CC, or another open license.
- Rewrite lines 29–32 in the abstract. The statement "The method was also used to predict all unambiguous VUSs in ClinVar" is unclear. VUSs are inherently uncertain, so this needs clarification.
- replace Figure 4 with a well-formatted table. The figure is difficult to read, and a table would be more effective.

Reviewer 2 Report

Comments and Suggestions for Authors

The authors develop a tool called PON-P3 to predict the pathogenicity of amino acid substitutions. This tool provides more accurate performance evaluation than existing prediction tools by comprehensively utilizing multiple features and machine learning algorithms. Before consider publishing in our journal, some revisions are needed.

 Comments:

1.       The authors have utilized datasets provided by previous works or online services to identify certain features, followed by some machine learning algorithms. Please discuss why your algorithm are better than other in the Discussion part.

 2.       The authors have only demonstrated the performance differences among various models at the 10-fold cross-validation stage, omitting the performance differences on a test set. It is crucial to include the performance disparities on a separate test set to provide a comprehensive evaluation of the models' generalizability.

 3.       When employing machine learning methods exclusively, the authors should conduct thorough ablation studies to demonstrate the validation process of their parameter search. This can provide evidence of the robustness and effectiveness of their chosen parameters and models.

 4.  The authors are requested to further elaborate on the innovations of PON-P3 compared to existing prediction tools, especially the improvements in prediction accuracy and computational efficiency.

 5.  The article mentions data collection from ClinVar and LOVD, but does not elaborate on the process of dataset construction. The authors are requested to provide more details, including data preprocessing, cleaning, and a statistical description of the final dataset.

 6.  The authors mention that 1,526 features were used and the 30 most important features were finally selected. Please describe the feature selection process in detail and explain why these features are important for prediction from a biological perspective.

7.  Could the authors please discuss the generalization ability of PON-P3 across different protein families and functional classes and provide the corresponding validation data?

Reviewer 3 Report

Comments and Suggestions for Authors

The manuscript introduces PON-P3, a novel machine learning-based tool for predicting the pathogenicity of amino acid substitutions. PON-P3 stands out due to its three-tier classification system (benign, pathogenic, and variants of uncertain significance [VUSs]) and its thorough integration of sequence, structural, and functional features. The authors conducted rigorous feature selection, model optimization, and comparative evaluation against 23 other predictors, demonstrating the tool's superior performance in realistic scenarios. While the work represents a significant advancement in the pathogenicity prediction of genetic variants, the manuscript requires further revisions and clarifications to strengthen its scientific rigor and presentation.

1. There are some grammatical errors in the manuscript. The author is advised to carefully review and make necessary corrections.

2. Several figures in the manuscript suffer from low resolution, which hinders their readability. High-resolution versions of all figures must be provided to ensure that the details, such as axes labels, legends, and critical visual components, are clearly visible.

3. Metrics like OPM (Overall Performance Measure) are complex and unfamiliar to some readers. Provide a clearer explanation of its components and how it advantages performance assessment over standard metrics like MCC or accuracy.

4. While the study discusses the proportion of VUSs predicted as pathogenic or benign, it lacks detailed analysis of misclassification rates for VUSs. Including: False positive/negative rates of VUSs and strategies for improving VUS classification accuracy in future iterations.

5. In order to enhance the credibility and scientific basis of the paper, I propose to cite some key bioinformatics literature in the paper, especially those related to miRNA detection, RNA interactions, or interpretable machine learning in bioinformatics, such as 10.1186/s13059-024-03357-w, 10.1109/TBDATA.2023.3334673, 10.1186/s12967-024-05372-8, 10.1007/s11432-024-4098-3, and so on.

Comments on the Quality of English Language

The English could be improved to more clearly express the research.

Round 2

Reviewer 1 Report

Comments and Suggestions for Authors

ad 1a
"We do not speculate" – and there is no need to.
In my opinion, there should simply be information indicating that the provided variant does not match the sequence found for the given UID in MANE 1.3. For example, S412N is valid, while A412N is invalid, as position 412 in NP_000008.1 is S. Since your tool is limited to sequences from MANE 1.3, such validation is trivial to implement and, in fact, necessary. Currently, MANE 1.4 is available, and over time, more new or obsolete records will appear. Therefore, it is essential to include information if a given UID is outside MANE 1.3 (e.g., NP_000008.2) or does not match its sequence.

ad 1b
While acceptable, based on my experience using your service, it appears that when the server is busy, the page will look like frozen for an extended time, leading many users to close it. I am unsure if such requests are processed when the user closes the tab prematurely. There should be a standard queuing system that takes user input from the web frontend, assigns a UID to the data, and immediately (within 1–2 seconds) notifies the user that their job has been added to the queue. The result can then be sent via email at a later time. Whether the wait is 1 minute, 1 hour, or 1 day due to input size is irrelevant. Additionally, providing a rough estimation of processing time at this point would be helpful.

ad 1d
There should be information in the subject of the email indicating whether the input_transcript or input_genomic subservice was used. Even if the output is the same, the input is different. Consider also adding a textbox for users to provide a submission title. Running your service for instance 20 times and trying to find a specific result in emails that all look the same is extremely frustrating.

ad 3
Please provide all the data, i.e., your predictions alongside the prediction files from other methods, ensuring they match. For example, create a folder for each training, testing, and validation dataset for each method so they can be processed one by one to produce the scores presented in Table 4. Include the scripts as well.
"We have not seen it necessary as these are third-party data and publicly available" – either provide all the data or withdraw the manuscript.

ad 5
"Within the provided time for revision (in the middle of the holiday season) and due to requests also from the other Reviewers, it was not possible to test another algorithm."
At this point, my patience has run out. Reject.

Author Response

ad 1a

"We do not speculate" – and there is no need to.

In my opinion, there should simply be information indicating that the provided variant does not match the sequence found for the given UID in MANE 1.3. For example, S412N is valid, while A412N is invalid, as position 412 in NP_000008.1 is S. Since your tool is limited to sequences from MANE 1.3, such validation is trivial to implement and, in fact, necessary. Currently, MANE 1.4 is available, and over time, more new or obsolete records will appear. Therefore, it is essential to include information if a given UID is outside MANE 1.3 (e.g., NP_000008.2) or does not match its sequence.

REPLY:
Those variants that PON-P3 cannot predict are listed in a separate section in the PON-P3 report. Clarification there indicates that most often, there is a mismatch with the used MANE reference sequence. We plan to update the reference sequences in the future. 

ad 1b

While acceptable, based on my experience using your service, it appears that when the server is busy, the page will look like frozen for an extended time, leading many users to close it. I am unsure if such requests are processed when the user closes the tab prematurely. There should be a standard queuing system that takes user input from the web frontend, assigns a UID to the data, and immediately (within 1–2 seconds) notifies the user that their job has been added to the queue. The result can then be sent via email at a later time. Whether the wait is 1 minute, 1 hour, or 1 day due to input size is irrelevant. Additionally, providing a rough estimation of processing time at this point would be helpful.

REPLY:
Thank you for this important observation. We have modified the web page to provide information about the submission immediately. Whether the tab is closed or not does not affect the submission once the submit button has been pressed. Once the query is completed, results are sent to the user via email. 

The processing time for jobs in PON-P3 depends on multiple factors, including the number and type of variants (protein, genomic, or transcript) and the server load. Since PON-P3 relies on third-party tools for converting genomic and transcript variations to protein-level data, providing a precise estimation of processing time is not feasible. 

ad 1d

There should be information in the subject of the email indicating whether the input_transcript or input_genomic subservice was used. Even if the output is the same, the input is different. Consider also adding a textbox for users to provide a submission title. Running your service for instance 20 times and trying to find a specific result in emails that all look the same is extremely frustrating.

REPLY:

Thank you for highlighting this valuable point. The report indicates the submitted variant and protein variant used for prediction. 

A textbox now allows users to provide a title for their submission. We hope these improvements enhance the user experience and make organising and identifying results easier.

ad 3

Please provide all the data, i.e., your predictions alongside the prediction files from other methods, ensuring they match. For example, create a folder for each training, testing, and validation dataset for each method so they can be processed one by one to produce the scores presented in Table 4. Include the scripts as well.

"We have not seen it necessary as these are third-party data and publicly available" – either provide all the data or withdraw the manuscript.

REPLY:

We appreciate the importance of transparency in research. We host VariBench, the only database for datasets for variation interpretation tool development and benchmarking. The PON-P3 webpage contains datasets for training and test data (these are also in VariBench) and predictions of all possible amino acid substitutions in human proteome (about 205 million variants). In response to the request, we provide data for all the compared methods used to produce the scores presented in Table 4. These data can be downloaded from the PON-P3 "About" page. 

ad 5

"Within the provided time for revision (in the middle of the holiday season) and due to requests also from the other Reviewers, it was not possible to test another algorithm."

At this point, my patience has run out. Reject.

REPLY: No comment about patience.

ABOUT ANOTHER ALGORITHM: We present in Table 2 an additional algorithm (SVM with RBF kernel) as initially requested by the Reviewer. The results of this algorithm are discussed in the text. The performance was lower than that of the gradient boosting algorithms.

Reviewer 2 Report

Comments and Suggestions for Authors

The authors have well replied my comments and the quality of the manuscript have been improved a lot. Thus, I recommend accept.

Author Response

Nothing to comment in here. The Reviewer suggests acceptance.

Round 3

Reviewer 1 Report

Comments and Suggestions for Authors

The authors have addressed the majority of my concerns. Thus, I find the revised version to be significantly improved and suitable for publication. Consequently, I recommend the manuscript to be accepted for publication.

Author Response

We thank for the comments and suggestion to publish.